# The Role of Plant-Based Diets and Personalized Nutrition in Endometriosis Management: A Review

**DOI:** 10.3390/medicina61071264

**Published:** 2025-07-13

**Authors:** Marijana Matek Sarić, Tamara Sorić, Ana Sarić, Emili Marušić, Miran Čoklo, Mladen Mavar, Marija Ljubičić, Nataša Lisica Šikić

**Affiliations:** 1Department of Health Studies, University of Zadar, Splitska 1, 23000 Zadar, Croatia; marsaric@unizd.hr (M.M.S.); mljubicic@unizd.hr (M.L.); nlisicasi@unizd.hr (N.L.Š.); 2Psychiatric Hospital Ugljan, Otočkih dragovoljaca 42, 23275 Ugljan, Croatia; ravnatelj@pbu.hr; 3School of Medicine, Catholic University of Croatia, Ilica 242, 10000 Zagreb, Croatia; asaric1@unicath.hr; 4Department of Pediatrics, General Hospital Zadar, Bože Peričića 5, 23000 Zadar, Croatia; emili.colak@gmail.com; 5Institute for Anthropological Research, Ljudevita Gaja 32, 10000 Zagreb, Croatia; mcoklo@inantro.hr

**Keywords:** endometriosis, plant-based diet, personalized nutrition, sustainable healthcare, holistic approach

## Abstract

Endometriosis is a chronic, estrogen-dependent inflammatory condition that affects multiple organ systems and significantly impairs the quality of life in women of reproductive age. While conventional hormonal therapies may alleviate symptoms of endometriosis, they are also frequently associated with intolerable side effects. As a result, there is growing interest in complementary, non-invasive strategies to support long-term disease management. This review explores the potential of plant-based diets and personalized nutrition as adjunctive approaches in endometriosis care. Plant-based dietary patterns, which are rich in antioxidants, phytochemicals, dietary fiber, and essential micronutrients, have been shown to reduce systemic inflammation, modulate estrogen activity, and alleviate pelvic pain. Additionally, the use of medicinal plants, such as curcumin and ginger, has demonstrated anti-inflammatory and anti-proliferative effects in preclinical studies. Moreover, identifying and addressing individual food sensitivities, particularly to gluten, dairy, or fermentable oligosaccharides, disaccharides, monosaccharides, and polyols, may improve gastrointestinal and inflammatory symptoms in susceptible individuals. Future research should focus on high-quality clinical trials and integrative care models to evaluate the long-term efficacy, safety, and sustainability of these individualized nutritional interventions in the holistic management of endometriosis.

## 1. Introduction

Endometriosis is a chronic, estrogen-dependent inflammatory disease that significantly impacts women’s physical, emotional, and social well-being [1,2]. It is characterized by the presence of endometrial tissue outside of the endometrium and myometrium, consisting of both endometrial glands and stroma [3]. This condition can lead to a range of debilitating symptoms, including chronic pelvic pain, heavy menstrual bleeding, and infertility [3,4].

Despite extensive research, there is still much to learn about the pathogenesis of endometriosis. Current treatment strategies, ranging from hormonal therapies to surgery, often provide only temporary relief and are associated with significant side effects [5,6]. Consequently, there is growing interest in complementary strategies that address the underlying inflammation and hormonal imbalances associated with the disease.

Emerging evidence suggests that dietary and lifestyle modifications, particularly the strategic incorporation of vitamins, minerals, and anti-inflammatory foods into the diet, may play a critical role in managing symptoms and improving quality of life [7]. Among these, the management of endometriosis through plant-based diets and personalized nutrition has gained increasing attention. Plant-based dietary patterns, which are naturally rich in anti-inflammatory compounds such as antioxidants, phytochemicals, and dietary fiber, have been associated with reduced systemic inflammation and pain in affected individuals [8]. Furthermore, the concept of personalized nutrition, which tailors dietary interventions to an individual’s unique genetic, metabolic, and lifestyle factors, offers a more targeted strategy for addressing the complex and heterogeneous nature of endometriosis [9,10].

Although several reviews have examined the relationship between nutrition and endometriosis, evidence specifically addressing the role of plant-based diets remains limited [8,10,11,12,13]. Heterogeneity in study designs and outcomes, as well as inconsistent definitions of plant-based diets, further complicates the interpretation of findings and limits firm conclusions about their efficacy [8,10]. Most of the existing literature has focused on broad dietary patterns or individual nutrients, with little attention to the potential benefits of a personalized nutritional approach in managing endometriosis. Therefore, while plant-based diets appear promising for reducing systemic inflammation and alleviating symptom severity, the current evidence is insufficient to support specific clinical guidelines for their use in this context [8,11]. These gaps highlight the need for a focused, comprehensive review that synthesizes emerging evidence on plant-based dietary interventions alongside personalized nutrition strategies in endometriosis.

Understanding the interplay between diet and endometriosis could open new avenues for non-invasive, sustainable symptom management with fewer side effects than conventional medical therapies. This paper will explore the role of plant-based diets in modulating inflammation and hormonal activity, and examine the potential of personalized nutrition in disease and symptom management.

## 2. Beyond the Pelvis: Endometriosis as a Systemic Disease

Endometriosis is traditionally classified as a chronic gynecological condition primarily affecting women of reproductive age [14]. However, recent advances in imaging, histopathology, and molecular biology have challenged the long-held perception of endometriosis as a localized gynecological disorder [15,16].

Ectopic endometrial lesions have been identified in distant locations such as the ovaries, intestines, bladder, lungs, diaphragm, and, in rare cases, even the central nervous system, including the brain [17,18,19,20]. This widespread distribution of lesions in various parts of the body demonstrates that endometriosis is a systemic disease affecting multiple organ systems [15,21].

In addition to the anatomical dissemination of lesions, systemic inflammation has been increasingly recognized as a hallmark of endometriosis [22]. Studies have consistently shown elevated levels of pro-inflammatory cytokines, chemokines, and growth factors in the peritoneal fluid and serum of women with endometriosis [23,24,25,26,27]. Furthermore, research suggests that immune system dysfunction is one of the key factors in the formation of endometrial lesions [28]. Immune cells, including neutrophils, macrophages, natural killer cells, and dendritic cells, appear to actively contribute to the angiogenesis, proliferation, and invasiveness of endometriotic tissue [28].

Moreover, endometriosis has been associated with a wide range of comorbidities, including autoimmune diseases (e.g., rheumatoid arthritis, systemic lupus erythematosus, and Hashimoto thyroiditis), metabolic disorders (e.g., insulin resistance and dyslipidemia), cardiovascular diseases, and several types of cancer (e.g., melanoma, ovarian, and breast cancer) [28,29,30,31].

Beyond physical comorbidities, the systemic nature of endometriosis extends to its profound psychological and neurological impacts. Women diagnosed with endometriosis frequently report symptoms such as chronic fatigue, cognitive impairment, anxiety, and depression [32,33,34,35,36].

Therefore, the systemic implications of endometriosis have important clinical and therapeutic consequences. Diagnosis based solely on pelvic symptoms or reproductive complaints may overlook non-gynecological manifestations [37]. In turn, treatment options targeting only the reproductive organs may inadequately address the systemic aspect of the disease [38]. That being said, a more holistic approach integrating gynecological, immunological, metabolic, and psychological dimensions is essential to properly treat this complex condition.

## 3. Plant-Based Diets and Their Benefits in Managing Endometriosis

Understanding endometriosis as a systemic disease strengthens the rationale for comprehensive management strategies, including dietary interventions. Anti-inflammatory and antioxidant-rich diets, such as plant-based dietary patterns, have been shown to be beneficial for women with endometriosis [8,10]. Recent research has indicated that adopting a plant-based diet may reduce systemic inflammation and lower circulating estrogen concentrations, offering relief from endometriosis-associated pain and hormonal imbalances [8,11,39,40]. The anti-inflammatory capacity of plant-based diets is already well-documented, and these positive effects observed in women with endometriosis are believed to be mediated by their high content of phytochemicals, dietary fiber, and essential micronutrients, including vitamins C and E [8,41,42]. In this context, the term “plant-based diet” encompasses a spectrum of dietary approaches that emphasize plant-derived foods, ranging from strictly vegan and vegetarian patterns to more flexible models such as the Mediterranean diet [43,44]. Table 1 presents key nutrients inherent in plant-based diets and their relevance to endometriosis management.

Additionally, plant-based diets typically minimize or exclude red meat and contain significantly lower levels of saturated and trans fatty acids compared to standard omnivorous diets [42,58]. A meta-analysis of observational studies found that higher consumption of red meat, saturated fats, and trans fats was associated with an increased risk of developing endometriosis, suggesting that reducing these dietary components may be beneficial for both prevention and management of the disease [59].

Similarly, one prospective study conducted in the United States of America reported that women who consumed more than two servings of red meat per day faced a 56% greater risk of developing endometriosis compared to those who ate one serving per week or less [60]. Conversely, no significant association was found between the consumption of poultry, fish, shellfish, or eggs and endometriosis risk [60]. These results align with an Italian study that also found a positive correlation between red meat intake and the likelihood of endometriosis [61].

The observed associations between red meat intake and endometriosis risk may be explained by several biological mechanisms. Consumption of red meat has been linked to elevated levels of estradiol and estrone sulfate, potentially contributing to increased steroid hormone concentrations, systemic inflammation, and the onset of endometriosis [8,62]. In addition, red meat intake may stimulate the production of pro-inflammatory molecules, which are believed to play a role in the initiation and progression of endometriotic lesions [8,63].

While plant-based diets appear to offer promising benefits in the management of endometriosis and may complement conventional medical therapies, it is essential to recognize that nutritional needs can vary considerably among individuals. As such, dietary interventions should be personalized rather than based on a one-size-fits-all approach in order to optimize outcomes and effectively address the complexity of the condition.

## 4. The Therapeutic Potential of Medicinal Plants in Endometriosis Management: An Integrated Approach

Beyond plant-based dietary patterns, the targeted use of medicinal plants and phytochemicals has garnered increasing interest as a complementary strategy for managing endometriosis [64]. Compounds such as curcumin and ginger have demonstrated anti-inflammatory, antioxidant, and anti-angiogenic properties, which may help alleviate symptoms associated with endometriosis [2,64,65,66]. However, the most critical limitation is the lack of human clinical trials. A major systematic review stated that curcumin clinical trials in endometriosis and ovarian illness are still scarce; thus, future studies need to be conducted to confirm the safety and efficacy of curcumin [67]. Despite promising preclinical evidence, no human clinical trials have specifically investigated ginger’s effects on endometriosis. The available clinical studies focus on primary dysmenorrhea rather than endometriosis-specific symptoms [68]. The lack of endometriosis-specific clinical trials represents a significant gap in the evidence base.

The mechanisms of action of the most studied phytotherapeutic agents in endometriosis management are outlined in Table 2.

Despite these encouraging findings, most current evidence stems from in vitro and animal studies, with human clinical research still in its early stages [2,66]. Therefore, high-quality randomized controlled trials are urgently needed to substantiate their efficacy and safety in clinical settings [2,75].

Integrating these bioactive compounds into broader plant-based dietary frameworks could potentially yield synergistic benefits [8]. This approach aligns with the growing interest in holistic management strategies that combine dietary interventions with conventional treatments, such as hormonal therapy [76,77,78,79]. By embracing a multifaceted treatment model, patients may experience enhanced symptom relief and an improved quality of life.

## 5. Dietary Triggers and Food Sensitivities in Endometriosis: Toward a Personalized Nutrition Approach

While not universally present, food sensitivities are common among women with endometriosis and may significantly influence symptom severity [80]. Therefore, tailoring nutrition plans to individual symptom profiles, nutritional status, and specific food responses is essential for effective symptom management [10]. A particular food or dietary component that provokes symptoms in one individual may have no impact, or may even confer benefits, in another, highlighting the interindividual variability in physiological responses [81]. Personalized nutrition, guided by a clinical nutritionist in cooperation with a clinician experienced in endometriosis treatment, can help identify dietary triggers while avoiding unnecessary restrictions that could lead to nutritional deficiencies.

Among the most commonly reported food sensitivities in women with endometriosis are those related to gluten, dairy products, and fermentable oligosaccharides, disaccharides, monosaccharides, and polyols (FODMAPs) [82,83].

### 5.1. Gluten

In relation to endometriosis, gluten sensitivity is often mentioned in the context of non-celiac gluten sensitivity (NCGS) and celiac disease. There is also an allergic reaction to gluten, which is a rare condition similar to other food allergies, but is far less important in patients with endometriosis [84].

While both NCGS and celiac disease involve a negative reaction to gluten, they differ in their underlying mechanisms. Celiac disease is an autoimmune disease where the body’s immune system attacks the small intestine in response to gluten, while NCGS is a chronic, functional digestive disorder triggered by gluten consumption, where individuals experience symptoms like digestive discomfort, bloating, and fatigue, even though they do not have celiac disease [85].

Some studies report a higher prevalence of celiac disease among women with endometriosis, and vice versa, suggesting a possible shared inflammatory or immune-mediated pathway. However, causality remains unproven, and the clinical relevance for non-celiac women is unclear [86,87].

Women with endometriosis are commonly advised to follow a gluten-free diet; however, the evidence regarding the effectiveness of this dietary strategy remains inconclusive. The impact of a gluten-free diet on endometriosis-related pain and quality of life was evaluated in a retrospective study involving 207 patients with chronic pelvic pain associated with endometriosis [88]. After 12 months of adhering to a gluten-free diet, 75% of patients reported a significant reduction in painful symptoms, while 25% reported no improvement [88]. Improvements in general health perception and social functioning were also noted [88]. A UK-based study found that women with endometriosis who followed a gluten-free diet for three months experienced a 50% reduction in pain [89]. In a dietary intervention study, women with endometriosis who reduced gluten and dairy intake (while increasing fruits, vegetables, and whole foods) reported less stress, improved sleep, greater energy, and overall better health. Those who did not adhere to the diet experienced more negative symptoms [90]. A large-scale survey from the University of Edinburgh indicated that 45% of respondents with endometriosis who adopted a gluten-free diet reported a reduction in pain, supporting findings from smaller clinical trials [81]. Additionally, individual case reports, such as a four-year follow-up of a woman with recurrent endometriosis, describe marked symptom improvement and even reduction in ovarian cyst size after strict adherence to a gluten-free diet [91]. Considering the reported symptom improvement in a substantial proportion of patients, a trial elimination of gluten might be a reasonable dietary approach to consider for women with endometriosis.

On the other hand, the effects of gluten consumption are highly susceptible to both placebo and nocebo responses [82]. Additionally, gastrointestinal symptoms observed in women with endometriosis may be attributable to comorbidities, such as irritable bowel syndrome (IBS), rather than being directly linked to endometriosis itself [82]. Recent reviews and large cohort studies (e.g., Nurses’ Health Study II) found little to no evidence that gluten intake is a strong factor in endometriosis etiology or symptomatology [82]. Another recent study concluded that current evidence does not support recommending a gluten-free diet for endometriosis, and such advice should be discouraged outside of diagnosed celiac disease [81]. Also, pro-gluten-free diet studies often lack control groups, randomization, or adequate blinding, and tend to rely on self-reported outcomes, introducing bias [84].

There is a notable lack of direct animal or in vitro studies specifically investigating the effects of gluten or a gluten-free diet on endometriosis development or progression. Most preclinical research focuses on dietary components such as antioxidants, omega-3 fatty acids, or general anti-inflammatory diets, rather than gluten specifically [81]. Also, it is generally known that effects observed in preclinical studies on dietary interventions often do not translate directly to clinical outcomes in humans, particularly when nutrients are consumed as part of a mixed diet rather than in isolation. Women with endometriosis are more likely to self-implement dietary changes, including a gluten-free diet, but often in conjunction with other modifications, making it difficult to isolate the effect of gluten exclusion [81].

Dietary restrictions, including adherence to a gluten-free diet, may also lead to inadequate intake of certain nutrients if not carefully planned and monitored by a clinical nutritionist. Long-term adherence to a gluten-free diet can reduce intake of dietary fiber, vitamins, and minerals, potentially leading to impaired diet quality and negative health outcomes [84]. Also, it is generally known that social and psychological burdens of strict dietary restriction may offset potential benefits, especially without clear evidence of efficacy.

Therefore, high-quality randomized controlled studies are needed to confirm the efficacy of this eating pattern in women with endometriosis and to establish evidence-based dietary recommendations.

### 5.2. Dairy Products

The relationship between dairy consumption and endometriosis is not yet fully understood.

Emerging evidence suggests that certain components of dairy products, primarily calcium and vitamin D, may have anti-inflammatory properties relevant to endometriosis management [92]. A prospective cohort study conducted in the United States of America found that women who consumed more than three servings of dairy products per day, as well as those with the greatest predicted vitamin D levels, were less likely to be diagnosed with endometriosis [93].

The most comprehensive meta-analysis examining dairy-endometriosis associations found that several specific dairy products, including whole milk, reduced-fat/skim milk, ice cream, and yogurt, showed no significant association with endometriosis risk [94]. Furthermore, when examining total milk consumption across five studies involving 2765 cases, no significant evidence was found that high milk intake was associated with the risk of endometriosis compared with low milk intake [94]. The evidence for low-fat dairy products similarly showed no significant protective effect. Three studies, including 2044 cases, indicated that high intake of low-fat dairy might not be associated with a reduced risk of endometriosis [94]. Additionally, the case-control study conducted in Italy between 1984 and 1999 found no association between milk or cheese consumption and endometriosis risk [61].

Despite these findings, clinical trials directly examining the impact of dairy products on symptom severity in women with endometriosis are lacking. Gastrointestinal discomfort related to dairy consumption has been frequently reported in this population, and reducing or eliminating dairy, one of the commonly employed dietary strategies, has shown effectiveness in alleviating symptoms experienced by women with endometriosis [83]. However, such gastrointestinal symptoms may also be attributable to coexisting conditions, such as IBS.

### 5.3. Fermentable Oligosaccharides, Disaccharides, Monosaccharides, and Polyols (FODMAPs)

FODMAPs are poorly absorbed short-chain carbohydrates known to induce gastrointestinal symptoms in individuals with IBS [95].

Women with endometriosis are at a significantly higher risk of developing IBS compared to women without endometriosis [96], and these two disorders frequently coexist. This association may be due to a shared immune system dysregulation or potential impacts of endometriosis treatments (particularly those involving oral contraceptives) [45].

The clinical presentation of endometriosis often overlaps with that of IBS, including symptoms such as abdominal pain, bloating, altered bowel habits, and food intolerances [97]. This significant overlap can complicate the diagnostic process, leading to misdiagnosis or delayed diagnosis of endometriosis [98].

Given the overlap in symptomatology, the low-FODMAP diet, commonly used in the management of IBS, has emerged as a potential therapeutic strategy for symptom relief in women with endometriosis [45], yet the evidence surrounding its efficiency remains inconclusive and conflicted. Although direct evidence in endometriosis populations remains limited, several studies suggest that women with endometriosis, particularly those with concurrent gastrointestinal symptoms, may benefit from FODMAP exclusion [83,96]. These findings support indications that adhering to a low-FODMAP diet may lead to improvements in endometriosis-related and gastrointestinal symptoms [82]. A prospective Dutch pilot study found that while participants adhering to dietary interventions reported less pain in four of six symptoms compared to baseline, when analyzed longitudinally against the control group, only bloating showed significant improvement (*p* = 0.049). Notably, this study found no significant influence of dietary strictness on reported pain scores, contrasting with previous retrospective findings [99].

One retrospective study found that 72% of women with both endometriosis and IBS showed > 50% improvement in bowel symptoms after four weeks of a low-FODMAP diet, compared to 49% in those with IBS alone [97]. However, this study has significant methodological limitations, including its retrospective design.

The most rigorous study to date is the 2025 EndoFOD randomized controlled crossover feeding study by Monash University, which demonstrated that 60% of participants responded to the low-FODMAP diet compared to 26% on the control diet (*p* = 0.008). This single-blinded trial showed 40% lower overall gastrointestinal symptom severity scores on the low-FODMAP diet compared to control [100].

The preclinical evidence base is notably sparse. A systematic review identified only one study specifically investigating low-FODMAP diet effects in endometriosis, which was retrospective and uncontrolled. The authors noted that prospective studies with low-FODMAP diets in patients with endometriosis with a clear description of their symptoms and comorbidities are needed to elucidate the therapeutic potential [101].

No animal or in vitro studies specifically investigating the low-FODMAP diet effects on endometriosis were identified in the literature. A systematic review examined 12 animal studies on dietary interventions in endometriosis, but none focused on FODMAP restriction. Animal studies have instead focused on other dietary interventions such as omega-3 supplementation, antioxidants, and caloric restriction [101].

However, in terms of patient compliance and treatment sustainability, it should be emphasized that long-term adherence to a low-FODMAP diet is not recommended due to potential nutritional deficiencies and negative impacts on the gut microbiota [102]. Recent evidence suggests complex interactions between gut microbiota and endometriosis that may confound dietary intervention outcomes. Studies show altered gut microbiota profiles in endometriosis, with increased Firmicutes/Bacteroidetes ratio and reduced overall microbial diversity. The low-FODMAP diet, while potentially beneficial for symptoms, may further alter gut microbiota composition with unknown long-term consequences [45].

Therefore, such dietary interventions should be implemented under the guidance of an experienced clinical nutritionist to ensure nutritional adequacy and individualized symptom management.

## 6. Sustainable Precision Nutrition in Endometriosis Management

The future of dietary interventions in endometriosis is evolving toward highly personalized, integrative strategies, fueled by breakthroughs in genetics, metabolomics, microbiome science, and digital health tools [9]. Personalized nutrition in this context is not merely about symptom control; it is a comprehensive approach that aims to optimize systemic health, address comorbidities, and promote long-term well-being through diet plans tailored to everyone’s unique biological, psychological, and lifestyle characteristics [103,104,105,106,107].

This personalized paradigm emphasizes the importance of understanding interindividual dietary responses, which can be influenced by an interplay of factors including genetics, epigenetics, gut microbiota, metabolic phenotype, and psychosocial stressors [9,108,109,110]. For example, while some individuals may benefit from high-fiber, plant-based diets, others with specific microbiome profiles or gastrointestinal sensitivities (e.g., IBS) might require modified approaches like the low-FODMAP diet [111]. Research indicated that specific bacterial populations are associated with endometriosis, suggesting that personalized dietary interventions might help restore a lost microbiome equilibrium [108].

Furthermore, advancements in genomics, metabolomics, and microbiome profiling are now allowing for more refined analyses of how individuals process and respond to different foods [111]. These technologies can help detect nutrient deficiencies, inflammatory biomarkers, hormonal imbalances, and microbial dysbiosis, all of which are relevant to endometriosis pathophysiology. For instance, metabolomic analyses can reveal disturbances in lipid metabolism or oxidative stress markers, which may then trigger specific dietary adjustments [112].

Mindful eating practices and dietary rhythm (e.g., meal timing and fasting patterns) are additional layers of personalization that can enhance metabolic and hormonal regulation. Anti-inflammatory diets often incorporate these practices alongside caloric balance, supporting both symptom relief and sustainable weight management when appropriate [10]. Among the most effective examples of such diets are the Mediterranean diet, veganism, and macrobiotic diets. The aforementioned diets offer holistic, sustainable, and health-supportive patterns well-suited for individuals with endometriosis [76]. The Mediterranean diet is particularly recognized for its anti-inflammatory properties, owing to its richness in antioxidants and beneficial fats, which can alleviate common endometriosis symptoms such as pelvic pain and hormonal imbalance [40]. Its diverse inclusion of vegetables, legumes, fruits, whole grains, nuts, and olive oil contributes to improved reproductive and metabolic health [113]. Similarly, vegan and macrobiotic diets, both emphasizing whole, minimally processed plant foods, are associated with lower intake of saturated fats and higher levels of dietary fiber, antioxidants, and phytonutrients, all beneficial for managing endometriosis [8].

Although social and cultural barriers may pose challenges to the widespread adoption of these diets, particularly in communities where animal products and processed foods are dietary norms, their evidence-based health benefits and sustainable attributes make them valuable options within personalized nutrition strategies for endometriosis care [114]. Research into synergies between sustainable and anti-inflammatory diets, particularly in younger populations, offers a promising pathway for cultivating lifelong healthy eating patterns that people would benefit from [92].

Despite these promising developments, several challenges and equity considerations remain. Implementing personalized and sustainable dietary strategies across diverse populations requires careful attention to nutritional adequacy, food accessibility, and cultural relevance [115]. Future interventions must therefore be designed with scalability, affordability, and inclusivity in mind. The practical implementation of individualized dietary interventions in clinical settings remains complex and multifaceted. One of the primary barriers to implementing personalized nutrition in endometriosis care is the lack of specialized training among healthcare professionals. While some specialized training programs exist, such as the “Endometriosis for Dietitians” course that equips practitioners with skills to support clients with endometriosis through nutrition, these programs remain limited in availability and accessibility. The course covers critical areas including immune system dysfunction, anti-inflammatory eating patterns, gut health management, and nervous system support, but requires significant time investment and specialized certification [116]. This knowledge gap is particularly problematic for endometriosis, where dietary interventions require understanding of complex interactions between inflammation, hormonal balance, and gastrointestinal symptoms. The multidisciplinary nature of endometriosis care necessitates collaboration between gynecologists, clinical nutritionists, and other healthcare professionals, but such integrated approaches are not widely established [114]. Patient adherence to personalized nutrition interventions represents a significant implementation challenge. Studies examining dietary interventions in endometriosis reveal substantial dropout rates and adherence issues. The Dutch pilot study found that 13 out of 47 patients (28%) withdrew before starting the diet, mostly due to lack of motivation [99]. This high dropout rate reflects the complexity and demanding nature of dietary modifications, particularly when patients must eliminate multiple food groups or follow restrictive eating patterns. Individualization requires extensive patient education, ongoing support, and frequent monitoring, which are resource-intensive and difficult to implement in standard clinical settings.

Economic barriers represent a significant obstacle to implementing personalized nutrition in endometriosis care. The cost of comprehensive nutritional interventions can be substantial, with estimates suggesting complex nutritional interventions cost USD 218.72 per patient during hospitalization, with additional costs of USD 814.27 when extended to community settings [117]. These costs often fall outside traditional healthcare coverage, creating financial barriers for patients seeking nutritional support.

Geographic accessibility also poses challenges, particularly in rural or underserved areas where clinical nutritionists with endometriosis expertise may not be available. While telehealth options exist, they may not provide the same level of comprehensive support as in-person consultations, particularly for complex cases requiring ongoing monitoring and adjustment of dietary interventions.

Digital health tools, such as mobile dietary tracking applications, telehealth consultations, and artificial intelligence-driven nutrition platforms, are emerging as valuable supports for both patients and clinicians. These tools can facilitate dietary monitoring, symptom tracking, and timely adjustments, enhancing both adherence and personalization [118].

Ultimately, the integration of personalized nutrition into endometriosis care represents a significant step toward precision medicine in women’s health. By aligning dietary strategies with individual needs, clinical insights, and environmental values, this approach has the potential to transform symptom management, empower patients, and promote sustainable health at both personal and societal levels.

## 7. Comparison with Previous Reviews: Shifting the Focus to Personalized Nutrition

The role of nutrition in endometriosis management has been evaluated in several recently published reviews [8,10,11,13,39,41,42,119,120]. These reviews were mainly focused on the potential role of diet and specific nutrients in the management of endometriosis, highlighting their potential benefits in alleviating symptoms and modulating disease progression. In contrast, this review places greater emphasis on the importance of an individualized nutritional approach, particularly the identification and management of personal food intolerances, such as sensitivities to gluten, dairy, and FODMAPs. This personalized perspective aligns with emerging concepts in precision nutrition and recognizes the heterogeneity of patient experiences and responses, thereby contributing a novel dimension to the evolving discourse on dietary interventions in endometriosis management.

## 8. Strengths and Limitations

This review is focused on plant-based diets and personalized nutrition as adjunctive strategies for managing endometriosis, highlighting several notable strengths. It addresses an important gap in conventional care by focusing on non-invasive, sustainable, and potentially low-risk interventions. By synthesizing evidence from domains such as phytotherapy, nutritional biochemistry, and gastrointestinal healing, the review provides a multidisciplinary perspective that supports more holistic, patient-centered models of care. A key strength is its emphasis on individualized approaches, including the identification of specific food sensitivities. This reflects a growing recognition of the heterogeneity of endometriosis and underscores the potential clinical relevance of tailored dietary interventions.

However, the review also has certain limitations. Much of the supporting evidence is derived from preclinical or observational studies, limiting the ability to draw firm conclusions regarding efficiency, optimal dosing, or safety in real-world settings. Furthermore, while the review discusses dietary strategies such as gluten and FODMAP exclusion, it does not clearly distinguish between well-substantiated recommendations and emerging hypotheses, which may reduce the practical utility of the guidance provided. The cited studies also exhibit considerable heterogeneity in study design, small sample sizes, risk of bias, and a general paucity of high-quality clinical trials, all of which contribute to the low overall quality of evidence. Taken together, these limitations underscore the need for cautious interpretation of the current findings and highlight the importance of further rigorous empirical validation to fully realize the clinical potential of these complementary strategies.

## 9. Conclusions

Endometriosis is increasingly recognized as a systemic inflammatory condition that necessitates a comprehensive, multidimensional, and patient-centered management approach. This review underscores the potential role of plant-based dietary patterns and personalized nutrition strategies in modulating the pathophysiological mechanisms underlying endometriosis. Diets rich in anti-inflammatory phytochemicals, dietary fiber, and antioxidants may contribute to the attenuation of chronic inflammation, oxidative stress, and estrogen-mediated pathways central to disease progression. Additionally, the adjunctive use of medicinal plants with evidence-based therapeutic properties presents a promising avenue for enhancing conventional medical therapies.

Emerging evidence supports the application of personalized nutrition, particularly the identification and elimination of individual dietary triggers such as gluten, dairy, and FODMAPs, as a targeted strategy to alleviate gastrointestinal and systemic symptoms, thereby improving patient quality of life. As the evidence base continues to evolve, future clinical practice guidelines should consider incorporating plant-centered, personalized nutritional interventions as a foundational element of integrative endometriosis management. Importantly, high-quality randomized controlled trials are needed to robustly validate the efficacy and safety of these dietary interventions and to inform evidence-based recommendations.

## Figures and Tables

**Table 1 medicina-61-01264-t001:** Key nutrients in plant-based diets and their potential role in endometriosis management.

Nutrient	Primary Plant-Based Source	Potential Role in Endometriosis Management	The Strength of Evidence
Dietary fiber	Whole grains, legumes, fruits, vegetables	Binding to excess estrogen in the digestive system, facilitating its excretion, and thereby potentially reducing circulating estrogen levels [8,11]	Observational/preclinical studies
Prebiotics	Whole grains, legumes, fruits, vegetables, seeds	Anti-inflammatory and antioxidative properties, positive effects on immune regulation and gut microbiome balance [45,46]	Observational/preclinical studies
Probiotics	Fermented soy products (tempeh, miso), naturally fermented pickles, sauerkraut, kimchi, kombucha	Anti-inflammatory properties, immunomodulatory effects, positive impact on gastrointestinal health [45,46]	Observational/preclinical studies
Omega-3 fatty acids	Flaxseeds, chia seeds, hemp seeds, walnuts, algae	Anti-inflammatory properties contributing to the reduction in inflammation and pelvic pain in endometriosis [47,48]	Preclinical studies/clinical trials
Antioxidants	Fruits, vegetables, nuts, seeds, vegetable oils	Reducing oxidative stress and enhancing antioxidative defense, amelioration of disease symptoms [8,11,49,50]	Preclinical studies/clinical trials
Vitamin D	Fortified foods (e.g., plant-based milk), mushrooms	Anti-inflammatory and antioxidative capacity, amelioration of disease symptoms [8,11,51]	Observational/preclinical studies/clinical trials
Magnesium	Green leafy vegetables, legumes, nuts, seeds, whole grains	Reducing the symptoms of dysmenorrhea [11,52]	Observational/preclinical studies/clinical trials (no targeted randomized controlled trials in endometriosis per se)
Polyphenols	Berries, green tea, dark chocolate, cruciferous vegetables, red wine	Anti-inflammatory and antioxidant capacity, negative effect on invasion, adhesion, and angiogenesis, positive effect on apoptosis, modulation of estrogen activity [11,47,53,54]	Mostly preclinical studies
Flavonoids	Fruits, vegetables, green tea, red wine	Impact on mechanisms of inflammation, cell proliferation, apoptosis, angiogenesis, and oxidative stress, prevention of endometriosis invasiveness [54,55]	Observational/preclinical studies
Iron	Legumes, fortified cereals, several types of cereals and vegetables	Prevention of iron deficiency [11,56]	Observational/preclinical studies
Zinc	Legumes, seeds, nuts, whole grains	Immune regulation factor, anti-inflammatory and antioxidative capacity [57]	Observational/preclinical studies

**Table 2 medicina-61-01264-t002:** Phytotherapeutic agents and their mechanisms in endometriosis management.

Medicinal Plant	Active Compound	Potential Role in Endometriosis Management
*Curcuma longa* (Turmeric)	Curcumin	Anti-inflammatory and antioxidant capacity, apoptosis, and angiogenesis in endometrial lesions, prevention of endometriosis invasiveness [54,69]
*Zingiber officinale* (Ginger)	Gingerols, shogaols	Anti-inflammatory and antioxidant capacity, anti-proliferative effect [70], pain reduction in dysmenorrhea [71]
*Vitex agnus-castus* (Vitex)	Iridoid glycosides, flavonoids, diterpenes, essential oils	Exhibits phytoestrogenic effects, modulates hormonal balance, reduces symptoms associated with endometriosis, may improve menstrual cycle regularity [66]
*Withania somnifera* (Ashwagandha)	Withanolides	Antioxidant and anti-inflammatory properties, may help reduce stress and regulate hormonal levels [66]
*Calendula officinalis* (Calendula)	Triterpenoid saponins, flavonoids, carotenoids, essential oils, phenolic acids	Anti-inflammatory, wound-healing, and antioxidant capacity [66]
*Boswellia serrata* (Indian frankincense)	Boswellic acids	Anti-inflammatory, analgesic, and anti-proliferative effects [72]
Pycnogenol	Procyanidins, catechin, epicatechin, phenolic acids	Anti-inflammatory and antioxidant capacity, reduces endometriosis-associated pain [73,74]

## Data Availability

No new data were created or analyzed in this study. Data sharing is not applicable to this article.

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
