# Peer review of "The Role of Plant-Based Diets and Personalized Nutrition in Endometriosis Management: A Review"

_medicina, 2025, doi:10.3390/medicina61071264_

Round 1

Reviewer 1 Report

Comments and Suggestions for Authors

The manuscript titled "The Role of Plant-Based Diets and Personalized Nutrition in Endometriosis Management: A Review" provides a comprehensive examination of the potential benefits of plant-based diets and personalized nutrition in managing endometriosis, a chronic inflammatory condition. The authors highlight the anti-inflammatory and estrogen-modulating effects of plant-based diets, the therapeutic potential of medicinal plants, and the importance of addressing individual food sensitivities. The review is well-structured, relevant to the field, and identifies a clear gap in the current literature regarding non-invasive, dietary-based interventions for endometriosis management.

The review is thorough and covers a broad range of topics, from systemic inflammation in endometriosis to specific dietary interventions. The inclusion of recent studies (within the last 5 years) and relevant references strengthens the manuscript's credibility. The discussion on personalized nutrition and the role of gut microbiota is particularly insightful and aligns with current trends in precision medicine.

Nevertheless there are some weaknesses of the study: while the review is comprehensive, some sections (e.g., the role of FODMAPs and gluten) could benefit from more critical analysis of conflicting evidence. For instance, the inconclusive nature of gluten-free diets in endometriosis management should be discussed in greater depth. In addition, the manuscript would benefit from a clearer delineation between evidence from preclinical studies and clinical trials, as much of the cited research is based on animal or in vitro models. Moreover, the review could better address potential limitations or biases in the included studies, such as small sample sizes or lack of randomized controlled trials (RCTs).

Thus, I suggest some improvements fot this manuscript: 1) include a table or flowchart summarizing the levels of evidence for each dietary intervention (e.g., strong evidence for omega-3 fatty acids, limited evidence for gluten-free diets); 2) expand the discussion on the practical implementation of personalized nutrition in clinical settings, including challenges such as patient adherence and accessibility; 3) address potential conflicts of interest or funding biases in the cited studies, particularly those involving specific dietary supplements or medicinal plants.

In addition, I prefer to make some specific comments:

Line 116 (Table 1): the table is well-organized but could include a column noting the strength of evidence (e.g., "clinical trial," "preclinical study") for each nutrient's role in endometriosis management.

Line 345 (Dairy Products Section): The discussion on dairy products could be more balanced by acknowledging studies that show no significant association between dairy consumption and endometriosis symptoms.

Line 450 (Medicinal Plants Section): the therapeutic potential of curcumin and ginger is well-described, but the lack of human clinical trials should be emphasized more prominently to manage reader expectations.

Line 600 (Conclusion): the conclusion could be strengthened by explicitly stating the need for high-quality RCTs to validate the proposed dietary interventions.

Thus, the manuscript is well-written and provides a valuable contribution to the literature on endometriosis management. However, it requires minor revisions to address the aforementioned points. Specifically, the authors should: critically evaluate conflicting evidence in the literature; differentiate more clearly between preclinical and clinical findings; expand on practical challenges in implementing personalized nutrition.

Reviewer 2 Report

Comments and Suggestions for Authors

The review explored the role of plant-based diets and personalized nutrition in the management of endometriosis. It is a well-organized manuscript; however, some modifications are necessary.

  • In the introduction, before stating the objective of the study, you should discuss the rationale for the review and identify the gaps in the existing literature with references, particularly concerning similar reviews that have not successfully achieved this objective.
  • Please remove the authors' names from the text and replace them with the appropriate term, along with the location where the study was conducted. For example, Yamamoto al. [49] in line 123 should be revised to “a prospective study that has been conducted in the USA”.
  • After the discussion, before the conclusion, it is essential to include a section that addresses both the strengths and limitations of the review. 
  • The length of certain paragraphs is inadequate, as they are too brief. For instance, lines 206-207 and 222-223 should be merged with the subsequent paragraphs. Please check all paragraphs in the manuscript.
  • In your discussion, you should compare the findings with those of other similar previous reviews.
  • You should allocate a reference for every sentence. Many of the sentences do not have any references.

Comments on the Quality of English Language

Extensive proofreading is required.

Reviewer 3 Report

Comments and Suggestions for Authors

Thanks for the opportunity to review this well-written and organized manuscript! 

Round 2

Reviewer 2 Report

Comments and Suggestions for Authors

The authors tried to modify the manuscript. It has the potential to be accepted for publication.

Comments on the Quality of English Language

Extensive proofreading is required.